# The prevalence of anxiety and related factors among primary and secondary school teachers in Hanoi, Vietnam, during the COVID-19 pandemic in 2020

Thuy Thi Thu Tran[1], Quynh Chi Ta[2]*, Son Thai Vu[1], Huong Thi Nguyen[1], Thao Thu Do[3], Anh Hoang Dang[4]

1 Department of Occupational Health and Safety, Hanoi University of Public Health, Hanoi, Vietnam, 2 The Center for Injury Policy and Prevention Research, Hanoi University of Public Health, Hanoi, Vietnam, 3 Monitoring and Evaluation office, University of North Carolina office in Vietnam, Hanoi, Vietnam, 4 Vietnam Education Union, Hanoi, Vietnam

* tqc@huph.edu.vn

**Data Availability Statement:** All relevant data are within the paper and its Supporting information files.

## Abstract

The working conditions for teachers in Vietnam were characterized by increased workload and pressure, burdening teachers' well-being. The study aims to investigate anxiety prevalence and identify some related factors among primary and secondary school teachers in Hanoi after the first COVID-19 outbreak in 2020. This paper analyzed data of 481 teachers working at ten primary and secondary schools in Hanoi city. Anxiety was measured using the anxiety component of the Depression, Anxiety, and Stress scale 42 items. Multivariable logistics regression was performed to examine anxiety-related factors using SPSS 20.0 at a significant level p less than 0.05. The prevalence of anxiety symptoms was 42.4% and similar between primary and secondary school teachers. More secondary teachers reported moderate to severe anxiety symptoms than primary teachers did (31.6% and 27.7%). Primary school teachers who felt discomfort with their supervisor's assessment, high responsibility for student safety, and ever thinking of leaving their current job were more likely to report anxiety symptoms (OR (95%CI) = 2.8 (1.2–6.5), 3.6 (1.0–12.8), and 2.6 (1.3–5.4), respectively). Meanwhile, the discomfort of caring for many students or problematic students, repetitive work, and disagreement with coworkers were risk factors of anxiety among secondary school teachers (OR (95%CI) = 2.6 (1.2–5.8), 3.2 (1.1–9.2), 3.4 (1.3–8.8), and 3.7 (1.1–12.6), respectively). In conclusion, the prevalence of teachers with anxiety symptoms is on the rise, caused by the characteristics of the job and professional relationships. Tailored support for teachers in different grades is necessary to improve and prevent teachers' anxiety.

## Introduction

The teaching profession can be highly stressful, and teachers suffer from many mental health problems [1–7], especially anxiety during the COVID-19 pandemic [1]. A scoping review in

**Funding:** The authors received no specific funding for this work.

**Competing interests:** The authors have declared that no competing interests exist.

2022 reported the prevalence of anxiety among teachers ranging from 38% to 41.2% [2], even higher during the COVID-19 outbreak to 49.3% [1]. A longitudinal study also reported a higher level of anxiety during COVID-19 compared with the period before the outbreak [3]. In addition, anxiety is highly correlated with other mental health problems such as burnout, stress, and depression [1,2,4–7], leading to emotional exhaustion and lack of personal accomplishment about work [8], job dissatisfaction, and intention to leave [2,9,10]. Hence, research and interventions to address teacher anxiety are urgently needed [11].

Several studies reported the factors associated with anxiety disorders among teachers, including demographic characteristics [7], years of experience and teaching job [12,13], lack of administrative support [11,13], job demand [14,15], job satisfaction/absenteeism [16], resilience/class size [17], interpersonal conflict and organizational constraints [18], social support [14,19], and communication [13]. More research is essential to understand what factors are vital in triggering anxiety symptoms among educators.

In Vietnam, psychological problems, especially teachers' anxiety disorders, have not received due attention, and the factors associated with these problems have not been clearly understood. Hanoi is a city with an education system facing the challenges of a rapidly increasing mechanical population with pressure on facilities and human resources, and a shortage of schools, classrooms, and qualified teachers [20]. Teachers in primary and secondary schools are subject to heavy workloads and tremendous pressure [21]. Their mental problems might be intensified with the outbreak of COVID-19 with mitigations to prevent disease transmission hindering routine academic activities, technical issues with new online teaching methods, more stress and workloads while performance quality had to be guaranteed [22]. Therefore, this paper aims to describe the percentage of anxiety among primary and secondary school teachers and explore some factors associated with this condition.

## Materials and methods

### Study design, time, and location

This paper's analysis was conducted in 2022 using secondary data from the nationwide cross-sectional study on the status of psychological stress and solutions to relieve psychological stress in teachers. The original study was conducted by the National Education Union of Vietnam in September 2020 [23].

### Study participants

This paper extracted and analyzed data from 481 teachers currently working in ten primary and secondary schools in Hanoi city. Data were extracted from the original study's dataset of 3320 teachers from kindergarten to high school levels, which were collected in September 2020. Cases with missing data on more than half of the anxiety subscale's items were not included in the analysis.

### Sample size and selection of participants

In the original study, 3320 teachers were recruited in a three-step cluster sampling procedure. Firstly, seven provinces/cities were selected to represent seven agroecological regions in Vietnam. Secondly, two to four districts were selected from each province. Thirdly, schools from kindergarten to high school levels were chosen so that the study could recruit 10% of eligible teachers from each province, and then all teachers in these schools were invited to join the survey. In Hanoi, 481 teachers participated in the survey came from ten

primary and secondary schools in four districts, namely Hoan Kiem, Ha Dong, Long Bien, and Cau Giay districts.

In order to determine the prevalence of anxiety among primary and secondary school teachers in this paper, we applied the formula to specify the population proportion with the anticipated proportion of anxiety as 0.49 [1], with an absolute precision of 5% of the true proportion at 95% confidence. The result showed that the minimum required number of teachers was 335. Therefore, data from 481 teachers in the original database meeting the inclusion criteria were extracted for the study.

## Measurements

**Anxiety component of the DASS-42.** This paper's outcome of interest was teachers' anxiety, measured by the self-administered Anxiety sub-scales of the Vietnamese version of the Depression, Anxiety, and Stress Scale (DASS-42). The anxiety subscale contained fourteen items of the DASS-42 (items 2, 4, 7, 9, 15, 19, 20, 23, 25, 28, 30, 36, 40, 41) measuring anxiety symptoms of autonomic arousal, skeletal muscle effects, situational anxiety, and subjective experience of anxious affect. Some examples of Anxiety items included "I was aware of dryness of my mouth," "I felt scared without any good reason," and "I was worried about situations in which I might panic and make a fool of myself," Each item was rated at 4 Likert level from 0 (did not apply to me at all) to 3 (applied to me very much, or most of the time). Anxiety score was the sum of fourteen items' scores [24]. DASS-42 has been validated among the Vietnamese community [25]. In this study, the DASS-42 Anxiety Cronbach's alpha was 0.89 indicating the high internal consistency of the test items [26].

Anxiety is categorized into No anxiety, mild, moderate, severe, or extremely severe anxiety if the score is less than 8, from 8–9, 10–14, 15–19, or equal to 20 and over, respectively. Participants had no anxiety if their score was lower than 8 [24].

**Self-reported questionnaire to measure independent variables.** A self-reported questionnaire was developed to collect data on five groups of variables associated with anxiety in the literature review. These five groups included personal characteristics, teachers' responsibility, work regulation and benefits, work characteristics/environment, and work relationships.

Participants' characteristics included age ($<$39/$\geq$39), education level (Intermediate/College/University/Postgraduate), marital status (married/other), average monthly income in VND (up to 10 million/ more than 10 million), workplace (primary/ secondary schools), and intention to leave a job (yes or unsure/definitely no).

The other four groups of variables were presented as participants' perspectives of their working conditions, rated as "Discomfort" or "No discomfort". Teacher responsibilities included inappropriate working hours/schedules, high requirements on student safety, student-related situations to handle, problematic students, large classes, non-academic work, and assignments unsuitable to the profession. Work regulation/benefits included lack of time for self-improvement, inadequate eating and resting time at school, unsatisfactory remuneration regime, being criticized or punished, being disciplined or deduction of salary, regular inspection, job evaluation and reward, and lack of promotion and development opportunities. Work and environment characteristics included repetitive work and noisy working environment. Work relationships included teachers' unfavorable relationships with colleagues, supervisors, students, and parents.

## Data collection procedure

The self-reported questionnaires were prepared and delivered to the target teachers. Permission from the school management board and participants' consent forms were obtained before

the data collection. Researchers went to each school to introduce the study, invite participants, and distribute the questionnaires to teachers. After one week, researchers revisited and collected the administered questionnaires from participating teachers.

### Data analysis

The data were extracted to an Excel file and then cleaned and analyzed by SPSS 20.0 software. Missing data were treated accordingly to variable types. For continuous, discrete quantitative variables (e.g., age), missing data were input by the average value of that variable calculated on the whole dataset. For categorical variables, missing data were filled in based on logical ties with other variables or the missing distribution according to the ratio of selected classifications in that variable. For Likert scale variables, if the number of questions without a response was less than half the total number of questions, the missing data were calculated with the following formula: the mean, calculated by the total number of questions, divided by the number of missing questions. Conversely, if the number of non-responded questions was more than half the number of questions, the case was considered missing and discarded from the final analysis.

Descriptive data were presented in counts and percentages for categorical data while the age variable was presented as mean, standard deviation (SD), and min/max. Multivariate logistic regressions with the Enter method at the significant level p less than 0.05 were performed to identify factors associated with anxiety, stratified for primary and secondary teachers separately. The "No discomfort" group with exposure to individual and work-related factors was the reference in the regression model. The Hosmer-Lemmeshow test was performed to determine the Goodness of fit of the regression model.

### Ethical consideration

The University of Public Health ethics committee approved the study protocol in decision No. 449/2021/YTCC- HD3 before data collection and analysis. All information, including personal information, was encrypted to ensure confidentiality. The authors of this paper had no access to identifying information of study participants.

## Results

### Participants characteristics

The final analysis included 481 subjects with an average age of 38.9 (SD: 8.2). Table 1 shows that 75.3% of participants had a university degree, and 12.3% of teachers finished higher education. Most (85.7%) respondents were married. Participants with an average monthly income of less than 10 million VND accounted for 73.8%. About two-thirds (66.9%) of the subjects had no intention to change their current job. 59.3% of participants were primary teachers. No missing data were found on study variables.

### Prevalence of anxiety symptoms among study participants

Among 481 teachers participating in the study, 42.4% had anxiety symptoms. Noticeably, moderate, severe, and extremely severe anxiety percentages were 17.3%, 6.7%, and 5.4%, respectively (Fig 1). The prevalence of anxiety among primary school teachers was slightly higher than that of secondary teachers. However, the percentage of participants with more severe symptoms of anxiety was higher in the secondary teacher group.

**Table 1. Demographic characteristics of study participants (N = 481).**

|  | N | % |
|---|---|---|
| **Age: Mean (SD); min—max** | 38.9 (8.2); 22–54 ||
| **Age (years)** |  |  |
| <39 | 211 | 43.9 |
| ≥39 | 270 | 56.1 |
| **Education** |  |  |
| Intermediate | 4 | 0.8 |
| College | 56 | 11.6 |
| University undergraduate | 362 | 75.3 |
| Postgraduate | 59 | 12.3 |
| **Marital status** |  |  |
| Married | 412 | 85.7 |
| Others | 69 | 14.3 |
| **Average monthly income (VND)** |  |  |
| ≤10 million | 355 | 73.8 |
| >10 million | 126 | 26.2 |
| **Workplace (school)** |  |  |
| Primary | 285 | 59.3 |
| Secondary | 196 | 40.7 |
| **Intention to change job** |  |  |
| Definitely No | 322 | 66.9 |
| Yes, or Not sure | 159 | 33.1 |
| **Total** | **481** | **100.0** |

## Factors related to the presence of anxiety symptoms among primary and secondary school teachers

Table 2 presents the results of multivariate logistics regression to identify significant associations between personal and work-related factors and anxiety status. The analyses were performed separately for primary and secondary teachers.

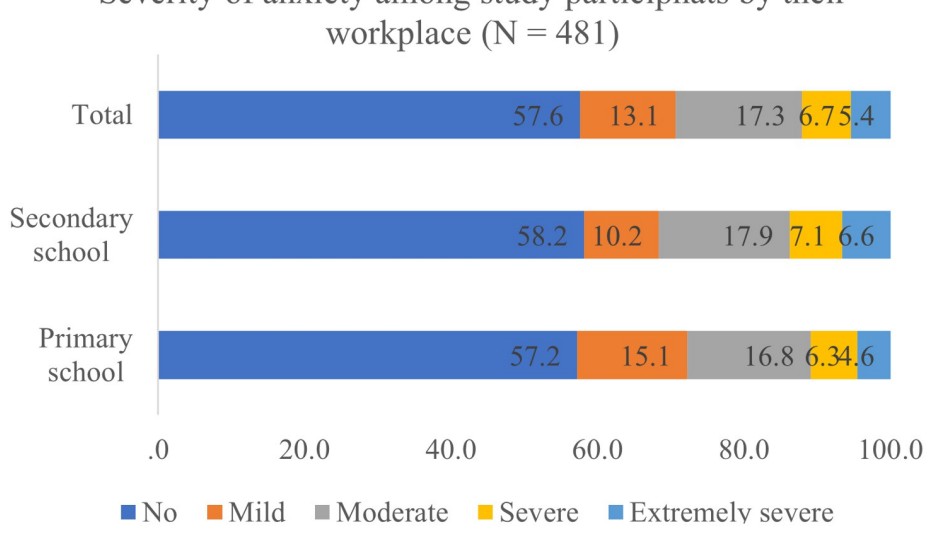

**Fig 1. Severity of anxiety among primary and secondary school teachers.**

**Table 2. Factors in association with anxiety status among primary and secondary teachers: Multivariate logistics regression.**

| Variables | | Primary teachers | | | | | | Secondary teachers | | | | | |
|---|---|---|---|---|---|---|---|---|---|---|---|---|---|
| | | Total (N = 285) | | Anxiety (n = 122) | | OR | 95% CI | | Total (N = 196) | | Anxiety (n = 82) | | OR | 95% CI | |
| | | N | % | n | % | | | | N | % | n | % | | | |

| Variables | | N | % | n | % | OR | 95% CI | | N | % | n | % | OR | 95% CI | |
|---|---|---|---|---|---|---|---|---|---|---|---|---|---|---|---|
| **Teacher responsibilities** | | | | | | | | | | | | | | | |
| Work hours | Not discomfort | 85 | 29.8 | 18 | 21.2 | ref | | | 52 | 26.5 | 9 | 17.3 | ref | | |
| | Discomfort | 200 | 70.2 | 104 | 52.0 | 1.3 | 0.6 | 2.9 | 144 | 73.5 | 73 | 50.7 | 1.7 | 0.5 | 5.5 |
| Unsuitable teaching assignment | Not discomfort | 176 | 61.8 | 64 | 36.4 | ref | | | 112 | 57.1 | 40 | 35.7 | ref | | |
| | Discomfort | 109 | 38.2 | 58 | 53.2 | 0.5 | 0.2 | 1.2 | 84 | 42.9 | 42 | 50.0 | 1.2 | 0.4 | 3.4 |
| High requirements for student safety | Not discomfort | 97 | 34.0 | 17 | 17.5 | ref | | | 64 | 32.7 | 10 | 15.6 | ref | | |
| | Discomfort | 188 | 66.0 | 105 | 55.9 | *2.8*[b] | *1.2* | *6.5* | 132 | 67.3 | 72 | 54.5 | 2.3 | 0.7 | 8.0 |
| A large number of students in charge | Not discomfort | 146 | 51.2 | 50 | 34.2 | ref | | | 96 | 49.0 | 24 | 25.0 | ref | | |
| | Discomfort | 139 | 48.8 | 72 | 51.8 | 1.0 | 0.5 | 2.0 | 100 | 51.0 | 58 | 58.0 | *2.6*[b] | *1.2* | *5.8* |
| Student-related situations to handle | Not discomfort | 120 | 42.1 | 37 | 30.8 | ref | | | 88 | 44.9 | 24 | 27.3 | ref | | |
| | Discomfort | 165 | 57.9 | 85 | 51.5 | 0.7 | 0.3 | 1.6 | 108 | 55.1 | 58 | 53.7 | 0.9 | 0.3 | 2.4 |
| Students with special problems | Not discomfort | 62 | 21.8 | 16 | 25.8 | ref | | | 52 | 26.5 | 10 | 19.2 | ref | | |
| | Discomfort | 223 | 78.2 | 106 | 47.5 | 1.1 | 0.5 | 2.7 | 144 | 73.5 | 72 | 50.0 | *3.2*[b] | *1.1* | *9.2* |
| Non-academic work | Not discomfort | 91 | 31.9 | 18 | 19.8 | ref | | | 75 | 38.3 | 19 | 25.3 | ref | | |
| | Discomfort | 194 | 68.1 | 104 | 53.6 | 1.4 | 0.6 | 3.4 | 121 | 61.7 | 63 | 52.1 | 0.9 | 0.3 | 2.5 |
| **Work regulation/benefit** | | | | | | | | | | | | | | | |
| Participation in training | Not discomfort | 163 | 57.2 | 51 | 31.3 | ref | | | 87 | 44.4 | 25 | 28.7 | ref | | |
| | Discomfort | 122 | 42.8 | 71 | 58.2 | 0.9 | 0.4 | 2.0 | 109 | 55.6 | 57 | 52.3 | 1.3 | 0.5 | 3.3 |
| Insufficient time for self-development | Not discomfort | 140 | 49.1 | 42 | 30.0 | ref | | | 58 | 29.6 | 15 | 25.9 | ref | | |
| | Discomfort | 145 | 50.9 | 80 | 55.2 | 0.9 | 0.4 | 1.8 | 138 | 70.4 | 67 | 48.6 | 0.5 | 0.2 | 1.5 |
| Inadequate eating and resting time at school | Not discomfort | 191 | 67.0 | 66 | 34.6 | ref | | | 122 | 62.2 | 41 | 33.6 | ref | | |
| | Discomfort | 94 | 33.0 | 56 | 59.6 | 1.0 | 0.5 | 2.2 | 74 | 37.8 | 41 | 55.4 | 0.5 | 0.2 | 1.4 |
| Unsatisfactory remuneration regime | Not discomfort | 184 | 64.6 | 63 | 34.2 | ref | | | 120 | 61.2 | 41 | 34.2 | ref | | |
| | Discomfort | 101 | 35.4 | 59 | 58.4 | 0.8 | 0.4 | 1.8 | 76 | 38.8 | 41 | 53.9 | 1.6 | 0.6 | 4.1 |
| Be criticized or punished | Not discomfort | 159 | 55.8 | 53 | 33.3 | Ref[a] | | | 83 | 42.3 | 24 | 28.9 | ref | | |
| | Discomfort | 126 | 44.2 | 69 | 54.8 | 1.1 | 0.5 | 2.6 | 113 | 57.7 | 58 | 51.3 | 0.8 | 0.3 | 2.2 |
| Be disciplined or Salary deduction | Not discomfort | 183 | 64.2 | 67 | 36.6 | ref | | | 116 | 59.2 | 38 | 32.8 | ref | | |
| | Discomfort | 102 | 35.8 | 55 | 53.9 | 0.6 | 0.2 | 1.5 | 80 | 40.8 | 44 | 55.0 | 2.2 | 0.9 | 5.4 |
| Lack of promotion and development opportunities | Not discomfort | 214 | 75.1 | 80 | 37.4 | ref | | | 117 | 59.7 | 40 | 34.2 | ref | | |
| | Discomfort | 71 | 24.9 | 42 | 59.2 | 0.8 | 0.4 | 1.7 | 79 | 40.3 | 42 | 53.2 | 0.4 | 0.2 | 1.0 |
| Regular inspection | Not discomfort | 110 | 38.6 | 24 | 21.8 | ref | | | 53 | 27.0 | 10 | 18.9 | ref | | |
| | Discomfort | 175 | 61.4 | 98 | 56.0 | 2.0 | 0.8 | 4.6 | 143 | 73.0 | 72 | 50.3 | 0.9 | 0.3 | 3.0 |
| Job evaluation and reward | Not discomfort | 202 | 70.9 | 69 | 34.2 | ref | | | 119 | 60.7 | 38 | 31.9 | ref | | |
| | Discomfort | 83 | 29.1 | 53 | 63.9 | 1.8 | 0.7 | 4.9 | 77 | 39.3 | 44 | 57.1 | 1.8 | 0.7 | 5.0 |
| **Work characteristics/ environment** | | | | | | | | | | | | | | | |
| Repetitive work | Not discomfort | 184 | 64.6 | 61 | 33.2 | | | | 93 | 47.4 | 22 | 23.7 | ref | | |
| | Discomfort | 101 | 35.4 | 61 | 60.4 | 1.4 | 0.7 | 2.9 | 103 | 52.6 | 60 | 58.3 | *3.4*[b] | *1.3* | *8.8* |
| Noisy work environment | Not discomfort | 70 | 24.6 | 20 | 28.6 | ref | | | 37 | 18.9 | 7 | 18.9 | ref | | |
| | Discomfort | 215 | 75.4 | 102 | 47.4 | 1.1 | 0.5 | 2.3 | 159 | 81.1 | 75 | 47.2 | 0.9 | 0.3 | 3.2 |
| **Work-related relationship** | | | | | | | | | | | | | | | |
| No support from coworkers | Not discomfort | 215 | 75.4 | 77 | 35.8 | ref | | | 126 | 64.3 | 46 | 36.5 | ref | | |
| | Discomfort | 70 | 24.6 | 45 | 64.3 | 1.1 | 0.4 | 3.2 | 70 | 35.7 | 36 | 51.4 | 0.8 | 0.3 | 2.3 |
| Disagreement with coworkers | Not discomfort | 212 | 74.4 | 76 | 35.8 | ref | | | 132 | 67.3 | 46 | 34.8 | ref | | |
| | Discomfort | 73 | 25.6 | 46 | 63.0 | 1.6 | 0.5 | 4.9 | 64 | 32.7 | 36 | 56.3 | *3.7*[b] | *1.1* | *12.6* |

*(Continued)*

**Table 2.** (Continued)

| Variables | | Primary teachers | | | | | | Secondary teachers | | | | | |
|---|---|---|---|---|---|---|---|---|---|---|---|---|---|
| | | Total (N = 285) | | Anxiety (n = 122) | | OR | 95% CI | | Total (N = 196) | | Anxiety (n = 82) | | OR | 95% CI | |
| | | N | % | n | % | | | | N | % | n | % | | | |
| Disagreement with supervisors | Not discomfort | 228 | 80.0 | 87 | 38.2 | ref | | | 141 | 71.9 | 53 | 37.6 | ref | | |
| | Discomfort | 57 | 20.0 | 35 | 61.4 | 0.4 | 0.1 | 1.6 | 55 | 28.1 | 29 | 52.7 | 0.3 | 0.1 | 1.3 |
| Supervisors' assessment | Not discomfort | 202 | 70.9 | 67 | 33.2 | ref | | | 119 | 60.7 | 41 | 34.5 | ref | | |
| | Discomfort | 83 | 29.1 | 55 | 66.3 | 3.6[b] | 1.0 | 12.8 | 77 | 39.3 | 41 | 53.2 | 0.9 | 0.3 | 3.2 |
| Parents' inappropriate response | Not discomfort | 127 | 44.6 | 32 | 25.2 | ref | | | 98 | 50.0 | 29 | 29.6 | ref | | |
| | Discomfort | 158 | 55.4 | 90 | 57.0 | 1.5 | 0.7 | 3.2 | 98 | 50.0 | 53 | 54.1 | 1.1 | 0.4 | 2.8 |
| Uncooperative parents | Not discomfort | 102 | 35.8 | 19 | 18.6 | ref | | | 66 | 33.7 | 17 | 25.8 | ref | | |
| | Discomfort | 183 | 64.2 | 103 | 56.3 | 2.2 | 0.9 | 5.2 | 130 | 66.3 | 65 | 50.0 | 1.6 | 0.6 | 4.3 |
| **Personal characteristics** | | | | | | | | | | | | | | | |
| Age | <39 yrs | 122 | 42.8 | 49 | 40.2 | ref | | | 89 | 45.4 | 48 | 53.9 | ref | | |
| | ≥39 yrs | 163 | 57.2 | 73 | 44.8 | 1.4 | 0.7 | 2.6 | 107 | 54.6 | 34 | 31.8 | 0.4 | 0.2 | 1.0 |
| Intention to leave | Definitely No | 211 | 74.0 | 79 | 37.4 | ref | | | 111 | 56.6 | 35 | 31.5 | ref | | |
| | Yes/ not sure | 74 | 26.0 | 43 | 58.1 | 2.6[b] | 1.3 | 5.4 | 85 | 43.4 | 47 | 55.3 | 2.1 | 0.9 | 4.9 |

[a] Reference group.

[b] $p < 0.05$.

For primary teachers, the discomfort with high requirements for student safety increased the odds of anxiety symptoms by 2.8 times. Participants who felt discomfort with supervisors' assessment were 3.6 times more likely to develop anxiety symptoms than those who did not have such negative feelings. Teachers with an uncertainty of staying with the job also reported a higher risk of anxiety (OR = 2.6, 95%CI = 1.3–5.4).

For secondary teachers, negative feelings about the number of students in charge and students with special problems were associated with teachers' increased anxiety (OR = 2.6, 95% CI = 1.2–5.8 and OR = 3.2, 95%CI = 1.1–9.2, respectively). Higher odds of anxiety symptoms also presented among secondary teachers who had discomfort with the repetitiveness of the job (OR = 3.4, 95%CI = 1.3–8.8) and disagreement with their coworkers (OR = 3.7 95% CI = 1.1–12.6), compared with teachers who had no discomfort with such working conditions.

## Discussion

The findings of our study show a high proportion of anxiety symptoms among primary and secondary teachers in Hanoi, especially at severe and extremely severe levels. Anxiety status is significantly associated with participants' intention to leave the job and several perceived work-related factors.

The proportion of participants with anxiety symptoms in our study was slightly higher than that of Agyapong's review [2], which reported the teachers' anxiety before COVID-19 ranging from 38.0% to 41.2%. Evidence of elevated anxiety among teachers during the disease outbreak in comparison with that of the prior period was demonstrated by Cortés-Álvarez (2022) in a longitudinal study [3]. Our result was also higher than that of all studies in Ozamiz-Etxebarria's review, with an anxiety rate from 9.5% to 37.2% in 2021 [27].

This paper's data were collected around September 2020, in the middle of the second COVID-19 wave in Vietnam. The nationwide spread of the pandemic, with increased new

cases and deaths every day, generated social fear and mental pressure on the whole country [28]. Different from other occupations which activities might be postponed because of the government's strict mitigation measures, education activities had to continue by alternative methods such as online lessons and social networking apps such as Zalo, Facebook, or YouTube [22]. Teachers faced many new challenges in teaching and supporting students to adapt to online learning, especially young primary and secondary students [22]. These working conditions not only contributed to the increased prevalence of teachers' anxiety but also intensified the severity of the problem, resulting in more teachers with severe anxiety symptoms. The high prevalence of teachers' anxiety in our study implies teachers' need for support, especially during the crisis to protect their mental health and work performance.

However, our findings were significantly lower than Santamaría's study in Spain (49.3%) during the COVID-19 pandemic in late 2020 [1]. This could be explained by the difference in the COVID-19 situation between Spain and Vietnam, which resulted in different impacts on teachers' anxiety. By September 2020, Vietnam reported 554 infections and 35 deaths, while the figures in Spain were about more than 200,000 new infections and 30,000 deaths [29]. In addition, at the same time in Spain, teachers' uneasiness and concerns about the risk of COVID-19 with the back-to-school policy might make them more anxious [1]. Interestingly, the percentage of teacher anxiety in our study was lower than that of some studies before COVID-19 in Malaysia (68%) [7] and Egypt (67.5%) [4]. Before the disease crisis, teachers had to worry about various aspects of work and life. Preparation for the Malaysian Certificate of Education resulted in increased job strain and unfavorable working conditions for teachers, causing the increased anxiety among Malay and Egyptian teachers [4,7]. Our finding emphasizes the importance of identifying contextual factors of anxiety in particular and mental problems in general for more appropriate solutions.

In our study, the subjects with anxiety symptoms at mild to moderate levels accounted for 30.4%, but 6.7% and 5.4% of participants suffered from severe and extremely severe anxiety symptoms. The proportion of subjects with severe anxiety symptoms in our study was similar to those with severe anxiety symptoms in the study in Egypt (7%). However, the survey results in Egypt had 19.7% of anxiety at a severe level, which could be explained by problems and challenges in the Egyptian education system at the time of the survey [4]. This disturbing condition lowered teachers' concentration on job performance and well-being [30]. Teachers need to be mentally strong to help students and work effectively. Hence, diagnosis of severe cases and interventions should be made to improve teachers' mental health.

Agyapong et al. in their review, reported that teachers' anxiety was significantly associated with their job responsibilities [2]. In our study, the high requirements of student safety had a significant effect on primary teachers (OR = 2.8, 95%CI = 1.2–6.5). Primary teachers are responsible for academic tasks and caring since students are transiting from kindergarten to the new educational system. For secondary teachers, supporting a large number of students who are in puberty with significant changes in physical and psychological conditions, especially problematic students, increases the risk of mental health problems including anxiety for teachers [31].

Unfavorable working conditions with high job demand and low social support are associated with poor mental health among teachers [7,11,14,15,18,19]. The repetitive aspect of the job, one feature of the work environment, is a component of job demand [14], which positively correlates with teacher anxiety. In our study, secondary school teachers who reported discomfort with the nature of repetitive work had a 3.4 times higher risk of anxiety than those without such uncomfortable feeling (95%CI: 1.3–8.8, p<0.05). In the educational professions, day-to-day routines of a teacher, especially of one in charge of a classroom, make teaching appear tedious and tiring [32]. Secondary school teachers in Vietnam, who were usually in charge of

one or two subjects, had to repeat their lessons for several classes. Teachers might feel the countless activities of their job demand are tedious, with writing lesson notes, grading exercises and homework, marking attendance, writing and filing student reports, and other administrative tasks.

Social support from colleagues, parents, and students plays a mediating role in teachers' anxiety, and lack of social support at work is strongly associated with their mental health disorders [14,15,19]. In our study, about one thirds of primary school teachers had discomfort with their supervisor's assessment (29.1%), but they had 3.6 times higher odds of developing anxiety in comparison with those who did not have such feelings. In addition, secondary teachers who were uncomfortable with the disagreement with coworkers presented a 3.7 times higher risk of developing anxiety symptoms than those without this feeling (OR = 3.7; 95%CI: 1.1–12.6). Previous studies also reported the association between work conflict and teachers' anxiety [32]. On the other hand, anxiety symptoms could hinder participants' ability to collaborate with others. Whether conflicts with coworkers caused teachers' anxiety or teachers' worsening mental state made them misevaluate other teachers' behaviors and cooperation, our study results show that developing a good relationship at work is essential and will be helpful to improve the teachers' psychological well-being. In their study, Hannon mentioned that emotional and coping training in early career might benefit teachers' continuing professional development [19] by helping them overcome such obstacles as relationships at work.

A significant association between primary teachers' anxiety and their intention to leave in our study is consistent with the literature [2,9,10]. Interestingly, teachers who were unsure of their intention had a higher risk of anxiety than those without the intention to leave (OR 2.6, 95%CI: 1.3–5.4). With the cross-sectional study design in our study, it is impossible to determine whether teachers' anxiety was caused by their intention to leave their education job or their deteriorated state made them want to change jobs. When teachers experience increased fear and worry about their work situations, especially during the COVID-19 pandemic in Vietnam, which was discussed above, they might lack enthusiasm for the work and, therefore, might think about not continuing their current teaching job. Conversely, teachers who already intend to leave their jobs might experience increased anxiety as they handle the strain and stress associated with teaching young people.

## Limitations

The study has several limitations. Firstly, a cross-sectional design can only present data at a particular time point, so the study cannot determine causal relationships. In addition, it is not possible to distinguish whether the anxiety symptoms present in the study subjects were long-standing or new cases. Secondly, this study did not explore coping methods used by teachers, so it was impossible to make recommendations on possible harmful coping mechanisms. Thirdly, the study only assessed anxiety symptom status based on the DASS-42 scale; no other health history or medical information that might affect the study results were collected. DASS-42 is only a screening tool that is unable to provide a formal diagnosis of anxiety. Personal or social values may influence an individual's feedback to self-reported questionnaires. In addition, several factors associated with anxiety in particular and mental health in general such as gender, stress and depression [1,2,7] were not included in the final analysis because of the inaccessibility of data. Future studies should have better control of such potential modifiers. Lastly, our results used secondary data and are representative only of primary and secondary teachers in urban districts of such a metropolis as Hanoi, the capital of Vietnam. The generalizability of study results to other groups of teachers in other places should be cautious. Despite its limitations, this study is one of the few that report the

prevalence of anxiety symptoms among primary and secondary teachers in Vietnam at the beginning of the COVID-19 outbreak.

## Conclusions

The percentage of anxiety symptoms among primary and secondary school teachers was 42.4%, in which 6.7% and 5.4% of teachers had symptoms at severe as extremely severe levels. Factors associated with anxiety among primary school teachers included the intention to change jobs, high requirements for student safety, and discomfort with supervisors' assessment. For secondary school teachers, the repetitive nature of work, class size, problematic students, and disagreement with coworkers were significantly related to increased odds of anxiety. Our findings imply the importance of regular anxiety screening for teachers to provide timely support to those in need. Stress management training would help teachers effectively deal with anxiety risk factors and improve their mental health. Future studies should consider more robust strategies such as longitudinal study design to identify factors causing teachers' anxiety in Vietnam accurately.

## Supporting information

**S1 Checklist. STROBE statement—Checklist of items that should be included in reports of *cross-sectional studies.***
(DOCX)

**S1 Data. Database.**
(SAV)

## Acknowledgments

The authors would like to thank the National Education Union of Vietnam for their permission to use and report data from their original study. We were thankful for the primary and secondary school teachers who participated and provided information.

## Author Contributions

**Conceptualization:** Thuy Thi Thu Tran, Quynh Chi Ta, Son Thai Vu, Huong Thi Nguyen, Thao Thu Do, Anh Hoang Dang.

**Data curation:** Quynh Chi Ta, Son Thai Vu, Thao Thu Do, Anh Hoang Dang.

**Formal analysis:** Thuy Thi Thu Tran, Son Thai Vu.

**Funding acquisition:** Anh Hoang Dang.

**Investigation:** Thuy Thi Thu Tran, Quynh Chi Ta, Son Thai Vu, Huong Thi Nguyen, Thao Thu Do, Anh Hoang Dang.

**Methodology:** Thuy Thi Thu Tran, Thao Thu Do, Anh Hoang Dang.

**Project administration:** Anh Hoang Dang.

**Resources:** Anh Hoang Dang.

**Supervision:** Thuy Thi Thu Tran, Anh Hoang Dang.

**Validation:** Thuy Thi Thu Tran.

**Writing – original draft:** Quynh Chi Ta, Son Thai Vu, Huong Thi Nguyen, Thao Thu Do.

**Writing – review & editing:** Thuy Thi Thu Tran, Anh Hoang Dang.

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
