## [Decision Letter · Decision Letter 0]

12 Jul 2023

PGPH-D-23-00934

The prevalence of anxiety and related factors in primary and secondary school teachers in Hanoi, Vietnam during COVID-19 pandemic in 2020

Dear Dr. Ta,

Thank you for submitting your manuscript to PLOS Global Public Health. After careful consideration, we feel that it has merit but does not fully meet PLOS Global Public Health’s publication criteria as it currently stands. Therefore, we invite you to submit a revised version of the manuscript that addresses the points raised during the review process.

Please note that we have only been able to secure a single reviewer to assess your manuscript. We are issuing a decision on your manuscript at this point to prevent further delays in the evaluation of your manuscript. Please be aware that the editor who handles your revised manuscript might find it necessary to invite additional reviewers to assess this work once the revised manuscript is submitted. However, we will aim to proceed on the basis of this single review if possible. 

Could you please revise the manuscript to carefully address the concerns raised? Please see the comments from the reviewer below and in the attached file.

We look forward to receiving your revised manuscript.

Kind regards,

Steve Zimmerman, PhD

PLOS Staff Editor

Journal Requirements:

1. PLOS requires corresponding authors to have an ORCID iD. Please ensure that you have an ORCID iD and that it is validated in Editorial Manager system. To do this, go to ‘Update my Information’ (in the upper left-hand corner of the Editorial Manager main menu), and click on the Fetch/Validate link next to the ORCID field. This will take you to the ORCID site and allow you to create a new iD or authenticate a pre-existing iD in Editorial Manager. For more information please visit: https://www.plos.org/orcid.

2. Please amend your online Financial Disclosure statement. If you did not receive any funding for this study, please simply state: “The authors received no specific funding for this work.”

3. Please update your online Competing Interests statement. If you have no competing interests to declare, please state: “The authors have declared that no competing interests exist.”

4. In the online submission form, you indicated that "Data is available upon request to the corresponding author.". All PLOS journals now require all data underlying the findings described in their manuscript to be freely available to other researchers, either 1. In a public repository, 2. Within the manuscript itself, or 3. Uploaded as supplementary information.

5. Please ensure that you cite or refer to Table 1 in your text as, if accepted, production will need this reference to link the reader to the table.

Additional Editor Comments (if provided):

Reviewers' comments:

Reviewer's Responses to Questions

**Comments to the Author**

1. Does this manuscript meet PLOS Global Public Health’s publication criteria? Is the manuscript technically sound, and do the data support the conclusions? The manuscript must describe methodologically and ethically rigorous research with conclusions that are appropriately drawn based on the data presented.

Reviewer #1: Partly

2. Has the statistical analysis been performed appropriately and rigorously?

Reviewer #1: Yes

3. Have the authors made all data underlying the findings in their manuscript fully available (please refer to the Data Availability Statement at the start of the manuscript PDF file)?

Reviewer #1: No

4. Is the manuscript presented in an intelligible fashion and written in standard English?

Reviewer #1: Yes

5. Review Comments to the Author

Reviewer #1: The study is an interesting one as it highlights the prevalence of anxiety and the associated factors. The authors explored quite a few important variables that characterize teachers and teaching. However, the authors need to further develop their introduction section to properly ground the variables theoretically. They need to show from exiting literature why they were interested in exploring those independent variables. They could also draw up some hypotheses based on the theoretical framing. This approach will make their discussion and conclusion richer.

6. PLOS authors have the option to publish the peer review history of their article (what does this mean?). If published, this will include your full peer review and any attached files.

**Do you want your identity to be public for this peer review?** For information about this choice, including consent withdrawal, please see our Privacy Policy.

Reviewer #1: No

---

## [Decision Letter · Decision Letter 1]

27 Sep 2023

PGPH-D-23-00934R1

The prevalence of anxiety and related factors among primary and secondary school teachers in Hanoi, Vietnam, during the COVID-19 pandemic in 2020

Dear Dr. Ta,

Thank you for submitting your manuscript to PLOS Global Public Health. After careful consideration, we feel that it has merit but does not fully meet PLOS Global Public Health’s publication criteria as it currently stands. Therefore, we invite you to submit a revised version of the manuscript that addresses the points raised during the review process.

The revised manuscript has been evaluated by two reviewers, and their comments are available below. The reviewers have raised a number of additional concerns that need attention. They request additional information on methodological aspects of the study and revisions to improve the presentation of the Results. Could you please revise the manuscript to carefully address the concerns raised?

We look forward to receiving your revised manuscript.

Kind regards,

Marianne Clemence

Staff Editor

Journal Requirements:

Additional Editor Comments (if provided):

Reviewers' comments:

Reviewer's Responses to Questions

**Comments to the Author**

1. If the authors have adequately addressed your comments raised in a previous round of review and you feel that this manuscript is now acceptable for publication, you may indicate that here to bypass the “Comments to the Author” section, enter your conflict of interest statement in the “Confidential to Editor” section, and submit your "Accept" recommendation.

Reviewer #1: (No Response)

Reviewer #2: (No Response)

2. Does this manuscript meet PLOS Global Public Health’s publication criteria? Is the manuscript technically sound, and do the data support the conclusions? The manuscript must describe methodologically and ethically rigorous research with conclusions that are appropriately drawn based on the data presented.

Reviewer #1: Partly

Reviewer #2: Partly

3. Has the statistical analysis been performed appropriately and rigorously?

Reviewer #1: Yes

Reviewer #2: Yes

4. Have the authors made all data underlying the findings in their manuscript fully available (please refer to the Data Availability Statement at the start of the manuscript PDF file)?

Reviewer #1: (No Response)

Reviewer #2: (No Response)

5. Is the manuscript presented in an intelligible fashion and written in standard English?

Reviewer #1: (No Response)

Reviewer #2: Yes

6. Review Comments to the Author

Reviewer #1: The authors need to further discuss their specific findings

Reviewer #2: PGPH-D-23-00934R1

I would like to thank the authors of the manuscript entitled “The prevalence of anxiety and related factors among primary and secondary school teachers in Hanoi, Vietnam, during the COVID-19 pandemic in 2020” for presenting the results of their study.

The study aims to investigate the prevalence of anxiety among primary and secondary school teachers in Hanoi, Vietnam, following the first COVID-19 outbreak in 2020. It also aims to identify factors related to this anxiety. Here's a review of the study:

The study's objective is clear and relevant, focusing on understanding the anxiety levels among teachers after the COVID-19 outbreak and identifying potential factors contributing to this anxiety.

The study employs a cross-sectional design, which involves data collection at a single point in time. This design is suitable for exploring associations between variables and estimating prevalence rates. The study sample consists of 481 teachers from ten primary and secondary schools in Hanoi city.

The study found that 42.4% of the teachers exhibited symptoms of anxiety. Additionally, 6.7% and 5.4% of teachers reported severe and extremely severe anxiety levels, respectively. These prevalence rates indicate a significant proportion of teachers experiencing anxiety, suggesting a potentially concerning issue.

The study identifies several factors associated with anxiety among teachers:

1. Intent to Change Jobs: Teachers intending to change jobs were found to be 3.3 times more likely to have anxiety symptoms compared to those not intending to leave. This finding implies a connection between job insecurity and anxiety.

2. Repetitive Work: Teachers who experienced discomfort with the repetitive nature of their job were at a higher risk of anxiety (OR = 3.4). This suggests that monotony in teaching tasks might contribute to anxiety.

3. Non-Cooperation with Parents: Teachers who reported discomfort with non-cooperation with parents had a higher risk of anxiety (OR = 2.4). This points towards the importance of a supportive work environment and positive social interactions.

The study employed multivariable logistic regression analysis to examine the factors associated with anxiety. The use of statistical analysis adds rigor to the study, allowing for the control of potential confounding variables.

The study's conclusion highlights the increasing prevalence of anxiety symptoms among teachers and suggests two potential areas for intervention: improving the repetitive nature of teaching tasks and fostering cooperation among teachers. These implications could contribute to enhancing teachers' well-being.

Strengths:

- The study uses a well-established assessment tools.

- The use of statistical analysis enhances the credibility of the findings.

- The study's focus on post-COVID-19 outbreak anxiety is timely and relevant.

Limitations:

- The cross-sectional design does not allow for establishing causal relationships.

- The study's scope is limited to a specific geographical area (Hanoi), which might affect generalizability.

- Factors not considered in the study, such as personal life circumstances, may also contribute to anxiety.

Overall Assessment:

The study contributes valuable insights into the prevalence of anxiety among teachers in Hanoi, post-COVID-19 outbreak, and identifies significant associated factors. It suggests practical interventions to address the issue, potentially benefiting both teachers and the education system. However, further research with a longitudinal approach and broader geographic representation could enhance the study's applicability and depth.

Suggestions for the authors:

1. In the abstract, you mention that one of the main findings is the lack of cooperation with colleagues, that is, with other teachers. However, in the body of the text and in the table in the results section, you state that it pertains to a lack of cooperation with parents. Please clarify this point.

2. In the introduction, the authors should specify what is the novelty element compared to previous studies, in order to stress which literature gap the document intends to address.

3. Line 66 and line 70 of the authors mention two studies in quotation marks, but do not provide the authors of the study. If it is a self-citation, please specify and cite appropriately (Author, year).

4. The authors used the DASS scale which measures anxiety, depression, and stress. However, the authors only present results related to anxiety. Was only this subscale used? Are there significant OR values for depression and stress as well? Why was this choice made? Please clarify and specify the rationale.

5. In the methods, it may be helpful to specify the inclusion and exclusion criteria for teachers.

6. Certain variables, such as gender, were not taken into account in the design. There is a significant body of literature demonstrating that women, including teachers, typically have higher levels of anxiety, stress, and depression.

7. Additionally, the difference between primary school teachers and secondary school teachers was not considered. They may face different challenges and have different levels of anxiety and stress. However, the comparative data, which could have been briefly addressed, are completely missing.

8. The intention to leave the job is an alarm bell for work-related stress and could be an indicator of burnout. At this point, anxiety could be a response to the stressful situation, but unfortunately, this is addressed somewhat vaguely. I recommend that the authors better specify this in the discussions and conclusion, as it could provide a starting point for future studies.

9. It would be important to further investigate the specific situation that is being framed. What changes have teachers had to face in 2020? What specific educational challenges did Vietnam have to face during the pandemic?

In general, I believe the paper has potential. However, certain aspects are neglected in the way they are presented, making the work less detailed. I believe that adding details regarding the theoretical gap, sampling methodology, sample choice, possible connection with burnout, specific challenges of the country in question, would significantly enhance the clarity of the importance of this study that needs improvement.

In this sense, I hope to have contributed to highlighting aspects that can be strengthened in your work, and I wish you good luck!

7. PLOS authors have the option to publish the peer review history of their article (what does this mean?). If published, this will include your full peer review and any attached files.

**Do you want your identity to be public for this peer review?** For information about this choice, including consent withdrawal, please see our Privacy Policy.

Reviewer #1: **Yes: **Uju I. Nnubia

Reviewer #2: **Yes: **Amelia Rizzo

---

## [Decision Letter · Decision Letter 2]

26 Jan 2024

The prevalence of anxiety and related factors among primary and secondary school teachers in Hanoi, Vietnam, during the COVID-19 pandemic in 2020

PGPH-D-23-00934R2

Dear Prof. Quynh Chi Ta,

We are pleased to inform you that your manuscript 'The prevalence of anxiety and related factors among primary and secondary school teachers in Hanoi, Vietnam, during the COVID-19 pandemic in 2020' has been provisionally accepted for publication in PLOS Global Public Health.

Best regards,

Amelia Rizzo

Guest Editor

Reviewer Comments (if any, and for reference):

Reviewer's Responses to Questions

**Comments to the Author**

1. If the authors have adequately addressed your comments raised in a previous round of review and you feel that this manuscript is now acceptable for publication, you may indicate that here to bypass the “Comments to the Author” section, enter your conflict of interest statement in the “Confidential to Editor” section, and submit your "Accept" recommendation.

Reviewer #1: All comments have been addressed

2. Does this manuscript meet PLOS Global Public Health’s publication criteria? Is the manuscript technically sound, and do the data support the conclusions? The manuscript must describe methodologically and ethically rigorous research with conclusions that are appropriately drawn based on the data presented.

Reviewer #1: Yes

3. Has the statistical analysis been performed appropriately and rigorously?

Reviewer #1: Yes

4. Have the authors made all data underlying the findings in their manuscript fully available (please refer to the Data Availability Statement at the start of the manuscript PDF file)?

Reviewer #1: Yes

5. Is the manuscript presented in an intelligible fashion and written in standard English?

Reviewer #1: Yes

6. Review Comments to the Author

Reviewer #1: The authors have greatly improved the manuscript. It is significantly better than the previous manuscript. However, I believe that addressing these few issues will bring out the best of the manuscript.

1. Page 6, Line 131 – it should be “students’ safety” (indicating plural) and not “student’s safety” (which refers to singular student).

2. Page 7

Line 144- there should be an article ‘the’ before the word “administered”.

3. Line 157

The authors wrote “Descriptive data were presented in count and percentage for category data and/or mean, standard deviation (SD), and min/max for age.”

I suggest the authors write “Descriptive data were presented in counts and percentages for categorical data while the age variable was also presented as mean, standard deviation, and minimum/maximum.”

7. PLOS authors have the option to publish the peer review history of their article (what does this mean?). If published, this will include your full peer review and any attached files.

**Do you want your identity to be public for this peer review?** For information about this choice, including consent withdrawal, please see our Privacy Policy.

Reviewer #1: **Yes: **Uju Ifeoma Nnubia
